# Improvement of Flowering Stage in *Japonica* Rice Variety Jiahe212 by Using CRISPR/Cas9 System

**DOI:** 10.3390/plants13152166

**Published:** 2024-08-05

**Authors:** Dengmei He, Ran Zhou, Chenbo Huang, Yanhui Li, Zequn Peng, Dian Li, Wenjing Duan, Nuan Huang, Liyong Cao, Shihua Cheng, Xiaodeng Zhan, Lianping Sun, Shiqiang Wang

**Affiliations:** 1College of Agronomy, Heilongjiang Bayi Agricultural University, Daqing 163711, China; he1323898@163.com; 2Chinese National Center for Rice Improvement and National Key Laboratory of Rice Biology and Breeding, China National Rice Research Institute, Hangzhou 311402, China; zhour0722@163.com (R.Z.); huangchenbo2022@163.com (C.H.); 13720149899@163.com (Z.P.); dyanli@foxmail.com (D.L.); 13140230176@163.com (W.D.); caoliyong@caas.cn (L.C.); chengshihua@caas.cn (S.C.); 3Baoqing Northern Rice Research Center, Northern Rice Research Center of China National Rice Research Institute, Baoqing 155600, China; 4Fuyuan Collaborative Breeding Innovation Center of China National Rice Research Institute, Jiamusi 156500, China; 5College of Chemical and Biological Engineering, Hechi University, Hechi 546300, China

**Keywords:** rice, flowering period, multiplex genome editing, breed improvement

## Abstract

The flowering period of rice significantly impacts variety adaptability and yield formation. Properly shortening the reproductive period of rice varieties can expand their ecological range without significant yield reduction. Targeted genome editing, like CRISPR/Cas9, is an ideal tool to fine-tune rice growth stages and boost yield synergistically. In this study, we developed a CRISPR/Cas9-mediated multiplex genome-editing vector containing five genes related to three traits, *Hd2*, *Ghd7*, and *DTH8* (flowering-stage genes), along with the recessive rice blast resistance gene *Pi21* and the aromatic gene *BADH2*. This vector was introduced into the high-quality *japonica* rice variety in Zhejiang province, Jiahe212 (JH212), resulting in 34 T_0_ plants with various effective mutations. Among the 17 mutant T_1_ lines, several displayed diverse flowering dates, but most exhibited undesirable agronomic traits. Notably, three homozygous mutant lines (JH-C15, JH-C18, and JH-C31) showed slightly earlier flowering dates without significant differences in yield-related traits compared to JH212. Through special Hyg and Cas marker selection of T_2_ plants, we identified seven, six, and two fragrant glutinous plants devoid of transgenic components. These single plants will serve as sib lines of JH212 and potential resources for breeding applications, including maintenance lines for *indica*–*japonica* interspecific three-line hybrid rice. In summary, our research lays the foundation for the creation of short-growth-period CMS (cytoplasmic male sterility, CMS) lines, and also provides materials and a theoretical basis for *indica*–*japonica* interspecific hybrid rice breeding with wider adaptability.

## 1. Introduction

Rice growth and yield are influenced by light–temperature interaction, affecting flowering, fruiting, and yield formation. The flowering period, a complex trait governed by multiple genes [1], is crucial in rice adaptation and yield optimization [2] by maximizing light and temperature resources [2]. Modifying the flowering period expands the cultivation area for high-quality rice varieties, addressing production challenges. Rice blast, caused by *Magnaporthe oryzae*, poses a significant threat to rice yield stability [3]. Developing disease-resistant rice varieties is pivotal for effective pathogen control.

Traditional rice breeding is time-consuming due to limited heritable variation and extensive phenotypic screening. Genetic engineering techniques, particularly CRISPR/Cas9, a cutting-edge gene-editing tool widely applied in crop genetic research to revolutionize trait improvement by precise DNA-level editing, accelerating breeding with enhanced efficiency and targeted outcomes. This approach offers rapid enhancements in flowering date, grain quality, stress resistance, and crop yield, minimizing time costs [4,5,6,7,8]. Noteworthy studies include editing the *DTH8* gene to alter flowering periods of *japonica* rice 99-25 and obtaining *DTH8* mutant material with an earlier flowering period [9]. In addition, this technology has been used to enhance resistant starch content through *SS3a* and *SS3b* editing, achieving a significant increase of 4.7–5.0% in single knockouts, and the double-knocked-out mutant exhibiting significant differences, with the resistant starch increasing 9.5–9.7%, creating new nutritious rice germplasm [10]. Wang Kejian’s group at the China National Rice Research Institute employed CRISPR/Cas9 gene-editing technology to knock out four reproduction-related genes, *PAIR1*, *REC8*, *OSD1*, and *MTL*, in the hybrid rice Chunyou84, achieving fusion-free reproduction of hybrid rice from 0 to 1, which is of great theoretical and practical significance [11]. Zhou Wenjia used CRISPR/Cas9 to edit the flowering period of the gene *Hd2* and the aroma gene *Badh2* of Suijing14 to shorten its flowering period [12]. Zhou employed CRISPR/Cas9 to edit broad-spectrum rice blast resistance genes *Bsr-d1*, *Pi21*, and *ERF922*. Single- or triple-mutants displayed high resistance to rice blast, with *Pi21* or *erf922* single mutants exhibiting enhanced resistance to leaf blight [13]. Plant gene-editing technology offers avenues for creating high-quality crop varieties, demonstrating the feasibility of directly editing genes in superior crop varieties for improved commercial crops. Rice flowering is controlled by multiple genes, with allele variants influencing regional adaptability under varying day lengths. *Hd2*, a core gene in rice photoperiodic flowering regulation systems, functions as a flowering repressor under long daylight conditions [14]. Its expression level correlates significantly with flowering time, regulating rice flowering by enhancing photoreceptivity [15]. *Ghd7*, a major QTL, controls grain number per spike, plant height, and flowering period in rice [16,17]. In the photoperiodic pathway under prolonged sunlight, *Ghd7* represses *Ehd1* expression, inhibiting flowering through the Ghd7-Ehd1-Hd3a/RFT1 pathway, and upregulates *OsCOL10* and *OsMFT1* as downstream flowering repressors [18,19]. Similarly, *DTH8*, a member of the rice HAP family (containing the HAP3D subunit), affects flowering, plant height, and grain number per spike [20] under extended sunlight, repressing flowering through protein interactions with *Hd1*, *Ghd7*, *DTH7*, and *HAPL1* [21].

Previous studies have shown that in the *Ehd1*-dependent photoperiodic flowering pathway, three flowering repressor genes, *Hd2*, *Ghd7*, and *DTH8*, negatively regulated rice varieties during the flowering period. Delaying rice flowering under prolonged sunlight increased plant height and the number of grains per spike, leading to higher yields. However, the extended reproductive period also increases the risk of encountering cold damage, cold waves, and lodging. Appropriately shortening the breeding period can prevent the above problems and expand rice cultivation to latitudes and altitudes that are currently unsuitable for cultivation due to the long planting cycle. *Hd2*, *Ghd7*, and *DTH8* have been reported to have slight effects on shortening the reproductive period of *japonica* rice variety. *Pi21* is a recessive rice blast resistance gene, and the *pi21* allele confers non-race-specific, durable resistance to *M. oryzae* [22]. The recessive version of the *Pi21* gene has a deletion of the first and second proline domains corresponding to 21 bp and 48 bp, respectively, associated with its resistance to rice blast [23]. *Pi21* negatively regulates disease resistance, and its loss-of-function shows resistance to rice blast. The use of this resistance is expected to overcome the problem of rice varieties prone to lose rice blast resistance.

Jiahe212 (JH212) is a traditional *japonica* rice variety renowned for its superior quality and serves as the maintainer line for Jiahe212A (JH212A), a high-quality hybrid line widely used in southern China. However, hybrid combinations involving JH212A often exhibit prolonged growth periods and low heat tolerance, hindering their widespread adoption. To address this, shortening the flowering period can mitigate heat stress during sowing, allowing for more flexible planting arrangements and facilitating the expansion of cultivation areas, especially northward. In this study, we employed the CRISPR/Cas9 system to conduct multi-gene editing targeting key genes in rice flowering regulation (*Hd2*, *Ghd7*, and *DTH8*), the rice blast resistance gene *Pi21*, and the flavor gene *Badh2* in JH212. The objective was to develop homozygous and stable mutant lines that enhance disease resistance and expedite the flowering period while preserving rice quality. These modifications aim to enhance breeding flexibility and promote wider adoption of high-quality *indica*–*japonica* interspecific hybrid rice varieties.

## 2. Results

### 2.1. Target Design and Vector Construction

The CRISPR-GE website (http://skl.scau.edu.cn/, accessed on 25 July 2021) was utilized for gRNA target site design. Primer sequences were selected based on high scores and proximity to the ATG start site. Target sites were designed on the first CDS1 of *Hd2*, *DTH8*, and *Pi21*, and the second CDS2 of *Ghd7* and *Badh2* for vector construction. The target site sequences included NGG (Figure 1). All five target sites were positioned close to the ATG start codon to facilitate the loss-of-function of the target gene. Finally, the rice genome was analyzed using BLAST through NCBI to confirm the specificity of the target sites.

### 2.2. Target Design and Off-Target Analysis

A set of candidate target sequences was generated by inputting these gene sequences into the target design tool on the CRISPR-GE website. Following the criteria for efficient target site selection, we opted for sequences with a GC content between 45% and 70%, off-target scores of less than 0.6, and sgRNA sequences with fewer than eight base pairs of complementarity. These selections were made to minimize off-target effects and enhance gene-editing efficiency. All five targets chosen for this experiment exhibited low off-target indices (Table 1), indicating the feasibility of gene editing at these sites. However, remaining vigilant about potential off-target risks during subsequent mutant screening is important.

### 2.3. Vector Construction and Genetic Transformation

In the laboratory of Kejian Wang at the China Rice Research Institute, the primary objective was to construct CRISPR/Cas9 expression vectors targeting five genes simultaneously using traditional homotetrameric enzyme technology [11]. Initially, the five target sequences were assembled into the intermediate vector SK gRNA. Subsequently, SK-gRNA-*Hd2*, SK-gRNA-*Ghd7*, SK-gRNA-*DTH8*, and SK-gRNA-*Pi21* were combined into a single intermediate vector named SK-gRNA-*Hd2*-*Ghd7*-*DTH8*-*Pi21*. Separately, SK-gRNA-*Badh2* was assembled into another intermediate vector, ligated with the intermediate vector containing four sgRNAs to form an intermediate vector encompassing five genes. Finally, the module with the five sgRNAs in tandem was integrated into the expression vector pCB2 using digestion–ligation methods. The resultant tandem module of five sgRNAs was further integrated into the expression vector pC1300-Cas9 via enzymatic cleavage and ligation. The constructed expression vector, CRISPR/Cas9-JH212, was subsequently sent to Wuhan Boyuan Company for genetic transformation of the *japonica* rice variety ‘Jiahe 212’ (Figure 2).

### 2.4. Acquisition of T_0_ Transgenic Positive Plants

The expression vector CRISPR/Cas9-JH212, carrying five targets, was employed to transform the recipient material JH212 via *Agrobacterium*-mediated methods, resulting in the generation of T_0_ transgenic lines. Detection of the vector region in the transgenic seedlings was conducted using vector-specific primers Cas9-F/Cas9-R to screen for transgene-positive lines. The analysis revealed that among 34 T_0_ transgenic seedlings, 30 were positive, corresponding to a positive rate of 88.23% (Figure 3).

### 2.5. Mutation Analysis of Transgenic Seedlings in T_0_ Generation

To analyze the mutation of *Hd2*, *Ghd7*, *DTH8*, *Pi21*, and *Badh2* genes in JH212, the upstream and downstream regions of the five target sequences were amplified using target detection primers. Subsequently, the PCR products were sequenced and analyzed. The sequencing results revealed mutations at the following frequencies: 26 positive plants were mutated at the *Hd2* target site position (mutation frequency of 86.66%), 25 positive seedlings were mutated at the *Ghd7* target site (mutation frequency of 83.33%), 25 positive seedlings were mutated at the *DTH8* target site (mutation frequency of 83.33%), and 26 positive seedlings were mutated at the *Pi21* target site (mutation frequency of 86.66%) (Table 2). The mutation genotypes included homozygous mutations, double allelic mutations, and various mutation types, such as base insertions and deletions.

### 2.6. Analysis of Mutation Sites in T_1_ Mutants

Based on the PCR sequencing results of the T_1_ plants, *Hd2* exhibited six types of homozygous mutations, including base insertions and deletions, notably insertions of T, A, C, and G between the third and fourth bases of PAM, and deletions of 4 bp and 46 bp. *Ghd7* displayed a total of eight types of homozygous mutations, including base insertions and deletions, notably insertions of T, A, and deletions of 3 bp, 4 bp, 5 bp, 7 bp, 28 bp, and 45 bp. *DTH8* showed four types of homozygous mutations involving deletions of 1 bp, 4 bp, and 6 bp. *Pi21* exhibited nine types of homozygous mutations, including insertions, deletions, and other base mutations, particularly insertions of single bases T, A, and G, as well as deletions of various bases (Figure 4).

### 2.7. Analysis of Mutation Sites in Homozygous T_1_ Mutants

After genotyping and analyzing the T_1_ generation mutant plants, 17 different genotypes were identified (Table 3), the flowering time of which was 24.25 days to 38.3 days earlier than that of JH212 (Table 3). Following phenotypic observations and agronomic trait investigations, three T_2_ lines were selected for the subsequent experiment: four-gene, three-gene, and two-gene homozygous and mutant lines. Subsequently, DNA sequences upstream and downstream of the target sites of the mutant plants were amplified using target detection primers (Table 3). Sequencing of the PCR products revealed that the mutations at the target sites of the T_1_ generation mutants were consistent with those observed in the T_0_ generation. This indicated that the genotypes of these three homozygous and mutant lines could be stably inherited. The three homozygous and mutant lines were established by self-pollinating the aforementioned three homozygous and mutant lines in the T_2_ generation, and they were named JH-C15, JH-C18, and JH-C31 (Table 3).

### 2.8. T_1_ Generation and T_2_ Generations of Cas9-Free Plants Were Obtained

To prevent ongoing gene editing by the Cas9 vector in transgenic plants, we implemented a strategy to remove the Cas9 vector starting from the T_0_ generation. This involved self-pollinating the transgenic plants across three consecutive generations. At the outset of each transgenic seedling generation, once we had confirmed gene editing in the transgenic lines, we conducted PCR amplification using two pairs of specific primers: Hyg-F/Hyg-R and Cas9-F/Cas9-R. If the genome of a transgenic plant amplified with both primer pairs did not yield the target fragment, we inferred that the target gene in that plant was mutated and the Cas9 vector was removed. These plants were then transplanted into the greenhouse for further cultivation. After two successive generations of self-pollination, we assessed the removal of the vector from the CRISPR/Cas9 knockout lines. While attempting to screen for plants without transgenic components in the T_1_ generation, we did not achieve homozygous mutant lines without transgene mutations due to the high temperature encountered during the T_0_ generation. Our focus was primarily on counting the removal of the Cas9 vector in the CRISPR/Cas9 knockout lines in the selected homozygotes lines in T_2_ generation, namely JH-C15, JH-C18, and JH-C31. The results indicated that out of 15 independent lines, we successfully obtained seven transgenic lines without the Cas9 vectors (Figure 5a,b). The three selected homozygotes lines showed significantly earlier flowering and maturity, significantly reduced plant height, but satisfactory performance in yield traits (Figure 5c–h).

### 2.9. qRT-PCR Detection of Gene Editing in Homozygous Mutant Lines

The expression levels of flowering and flowering genes *Hd3a*, *Hd2*, *Ghd7*, *DTH8*, and *Pi21* were assessed using qRT-PCR, and in the three homozygous and mutant lines JH-C15, JH-C18, and JH-C31 mentioned previously. The expression of *Hd3a* was significantly higher in the JH-C15, JH-C18, and JH-C31 homozygous mutant lines compared to the wild-type. Conversely, *Hd2* was significantly lower in all three homozygous and mutant lines compared to the wild-type. *Ghd7* expression was significantly lower in the JH-C18 four-gene homozygous and mutant line compared with the wild-type. *DTH8* expression was significantly reduced in the JH-C15 three-gene homozygous and mutant line and the JH-C18 four-gene homozygous and mutant line compared to the wild-type. Additionally, the expression of *Pi21* was significantly lower in all three homozygous mutant lines compared to the wild-type. These findings suggest that the transcript levels of *Hd2*, *Ghd7*, *DTH8*, and *Pi21* may have been affected and partially degraded in vivo following target editing by the Cas9 system, resulting in a significant downregulation of their expression in the mutant lines JH-C15, JH-C18, and JH-C31 (Figure 6a–c).

### 2.10. Phenotypic Analysis of T_2_ Generation Knockout Mutants

In this study, we investigated the phenotypes of the CRISPR/Cas9 knockout lines, focusing on flowering and yield-related traits in the three pure mutant lines mentioned earlier. JH-C15 showed wider grain (Figure 7a–e); JH-C18 exhibited larger grain length/width without significant influence on grain weight (Figure 7a–e); JH-C31 displayed wider grains without significant influence on grain weight (Figure 7a–e). All three lines exhibited significantly reduced plant height compared to JH212 (Figure 7f). The number of effective panicles significantly impacts the yield of a single plant, such as JH-C15. The main reason for the decrease in its yield is the significant decrease in the number of effective panicles (Figure 7g,h). The yield of JH-C18 and JH-C31 showed no significant decrease (Figure 7g,h), indicating their potential utilization value.

Among the identified flowering repressors, *Hd2* encodes a PRR protein homologous to *Arabidopsis TOC1*, *Ghd7* is a CO-like protein containing a CCT structural domain that controls plant height, flowering period, and spike number in rice, while *DTH8* serves as a pleiotropic regulator of plant height and flowering period. Ehd1 promotes the expression of *Hd3a* and *RFT1* under both long and short sunlight conditions, with *Hd2*, *Ghd7*, and *DTH8* acting as inhibitors of *Ehd1*. In the JH-C15 triple mutant strain, we observed a base T insertion at the *Hd2* target site and a base G deletion in the *DTH8* target site, resulting in premature protein translation termination and loss of gene function (Appendix A). Consequently, under long sunlight conditions, the JH-C15 mutant exhibited significantly earlier flowering compared to the wild-type, indicating successful knockout of the *DTH8* in JH-C15 and JH-C18 strains. Similarly, in the JH-C18 four-mutant line, a 46 bp deletion was observed at the *Hd2* target site, along with a base T insertion at the *Ghd7* target site and a four-base deletion at the *DTH8* gene sequence target site (Appendix A). This resulted in significantly earlier flowering in the JH-C18 mutant compared to the wild-type, confirming successful knockout of the *Hd2*, *Ghd7*, and *DTH8* genes in JH-C18. In the JH-C31 double mutant strain, a 4 bp deletion in the *Hd2* gene led to earlier spike flowering compared to the wild-type, indicating successful knockout of the *Hd2* gene (Figure 5g, Appendix A). Among the three families obtained, JH-C31 had fewer mutated genes and showed better performance in yield traits, effective tillering, yield per plant, grain length, and width ratio relative to the other two families, while JH-C15 in turn showed lower yield, especially in the decreased panicle number per plant compared to JH-C18 and JH-C31. This result may be attributed to the different number of genes mutated in these three lines, the different ways the genes were mutated, and the different magnitude of the effect of these three spike-stage genes. Hence, they showed various yield-related traits.

Rice blast fungus has different infestation and pathogenicity to different varieties, showing obvious physiological differentiation, and there are 128 physiological races (effective single-spore strains of fusarium oxysporum) of rice blast fungus in China. Previous studies have shown that the editing and transformation of the *pi21* gene in the *japonica* rice variety Nipponbare (NPB) could obtained new edited plants with enhanced resistance to the physiological race of *M. oryzae* RB22 [23]. In testing for rice blast resistance, we inoculated JH212 and three mutant lines at the three-leaf one-heart period with two strains of rice blast (PB22 and 17-4) conserved in our research group. Surprisingly, there was no difference in their disease resistance and no phenotype after inoculation of the edited lines with JH212. However, the selected physiological races used in this study exhibited high resistance in JH212 and the edited strains, resulting in a phenotype lacking disease resistance following rice blast inoculation (Appendix A).

## 3. Discussion

Factors like spiking duration, the disease resistance of specific physiological race varieties, and cropping schedules influence rice yield and the scope of planting. While extending the spiking period is commonly seen as a method to boost yields, there is growing interest in short-duration high-quality rice varieties due to their flexible sowing nature, which allows for expansion without significant yield loss. The spiking period, a critical physiological phase determining rice variety adaptability [24], has garnered attention from breeders. Improving the flowering period of existing *japonica* varieties can expand high-quality rice cultivation and address flowering period-related challenges in popularized varieties. Maximizing rice yield relies on cultivating varieties with optimal flowering periods for the planting region. Gene editing offers a rapid and stable breeding pathway favored by experts [25], boasting high efficiency, stability, and speed compared to traditional breeding methods like hybridization and backcrossing. CRISPR/Cas9 gene-editing technology has rapidly advanced in recent years, and it is extensively applied in creating germplasm resources [26], genetic enhancement [27], fertility improvement [28], disease resistance [13], and quality improvement of crops [28]. Its adoption in rice research has become instrumental in unraveling gene functions and molecular mechanisms [25].

In this study, we employed CRISPR/Cas9 gene-editing technology to modify *Hd2*, *Ghd7*, and *DTH8* in the rice flowering pathway and rice blast resistance gene *Pi21*. We successfully created new allelic variants in the JH212 background, establishing novel breeding materials for the flowering variation. In the T_0_ generation, 88.23% efficiency was achieved, with 30 out of 34 transgenic seedlings showing positive results. Sequenced analyses revealed high mutation frequencies as follows: *Hd2* (86.6%), *Ghd7* (83.3%), *DTH8* (83.3%), and *Pi21* (86.6%), with these predominantly involving base insertions and deletions. Our results align with previous studies. Li et al. achieved 77.8% simultaneous editing of *Hd2*, *Ghd7*, and *DTH8* across seven cultivars [29]; Yang et al. obtained an 86.7% mutation rate for *Pi21*; Xu et al. reported mutation frequencies of 75% for *Pita*, 85% for *Pi21*, and 65% for *ERF922* [30]. These data fully proved the effectiveness of our new-constructed vector.

Multiple genes or gene families jointly regulate important agronomic traits in rice. Using the CRISPR/Cas9 multiple knockout system, multiple genes can be targeted simultaneously to obtain plants with variable characteristics and to investigate the interactions among genes [31], offering new materials for crop breeding. Many new materials and germplasm for growth-targeted improvements have been obtained by the CRISPR/Cas9 multiple gene-editing system in cereal crops. Gene editing of maize flowering-related genes ZmCCT10, ZmCCT9, and ZmGhd7, with KN5585, CML312SR, LCL-1, and LCL-2 as stabilizing transformation receptors, yielded maize photoperiodic blunted sensing materials carrying single, double, and triple mutations [32]. While knockout of *ZmCCT10*, a homologue of the rice photoperiod-sensitive gene *Ghd7*, significantly influenced the flowering stage, the ZmCCT10-1IT and ZmCCT10-1IA lines showed earlier flowering and significant dwarfing [33]. Zhang et al. successfully created new maize varieties with fragrant rice flavor in maize seeds by constructing CRISPR/Cas9 co-knockout vectors for the *ZmBADH2-1* and *ZmBADH2-2* flavor genes [34]. Wang et al. used CRISPR/Cas9 technology to create wheat varieties with broad-spectrum resistance to powdery mildew by targeted mutagenesis of the powdery mildew susceptibility gene MLO [35]. In this study, we found that the probability of single gene mutation was minimal, and many of the mutations occurred at the same time, primarily as polygenic mutations. Notably, we identified eight homozygous mutation types for *Ghd7*, six homozygous mutation types for *Hd2*, nine homozygous mutation types for *Pi21*, and four homozygous mutation types for *DTH8*. These results are consistent with previous research. Yang et al. utilized CRISPR/Cas9 to simultaneously knock down eight genes highly expressed in the glutenin gene family and obtained seven mutants with varying combinations of mutations without affecting the content of storage substances and rice appearance, while the rice protein content was downregulated at different degrees [36]. Shen et al. successfully developed co-knockout vectors for eight genes associated with agronomic traits in rice using CRISPR/Cas9 and obtained homozygous six-mutant, seven-mutant, and eight-mutant strains [37]. This approach provides a strategy for rapidly introducing genetic diversity in the development of crop breeding. In constructing knockout lines, we have obtained materials to study further the dosage effect of alleles on phenotypes and genetic interactions between non-alleles, but also expanded the possibilities for the use of CRISPR/Cas9 gene-editing technology in targeted editing of the genomes of rice varieties, to obtain polymerization of favorable alleles. In future work, we can better utilize these haplotypes to enhance the diversity of flowering times. Moreover, by integrating several genes with flowering-promoting impacts, it is possible to breed varieties with very early flowering times.

In the 17 homozygous mutant lines obtained, the four-gene homozygous mutant line JH-C18, the three-gene homozygous mutant line JH-C15, and the two-gene homozygous mutant line JH-C31 were chosen for the following experiments by observing phenotypes and the investigation of agronomic traits. In analyzing the expression of flowering-related genes in the gene-edited homozygous lines, the expression of *Hd3a* in the homozygous mutant lines JH-C15, JH-C18, and JH-C31 was significantly higher than that of JH212. The transcript levels of *Hd2*, *Ghd7*, and *DTH8* might be impacted after being target-edited by the Cas9 system, partially degrading the RNAs in vivo, producing a significant downregulation of their gene expression in JH-C15, JH-C18, and JH-C31. Previous studies reported that in the *Ehd1*-dependent photoperiodic flowering pathway *Hd2*, *Ghd7*, and *DTH8* are the primary genes negatively regulated by flowering in rice varieties, delaying rice flowering under long sunlight conditions. Li et al. designed specific targets for three flowering genes, including *Hd2*, *Ghd7*, and *DTH8*, showing that the knockout progeny could reduce the flowering period to varying degrees [29]. Zhou et al. employed gene-editing technology to target the flavor gene *Badh2* and the flowering period gene *Hd2* to obtain improved rice varieties with improved flavor and early maturity [12]. Multi-target editing of *Pi21* and the flavor suppressor gene *OsBadh2* resulted in highly significant reductions in expression relative to the wild-type, as well as improved rice blast resistance and accumulation of 2-AP. In this study, the knockdown of *Hd2*, *Ghd7*, and *DTH8* resulted in significant advancement in the flowering period of mutant lines compared to the wild-type, indicating that the CRISPR/Cas9-mediated gene editing at the flowering period can achieve targeted fertility improvement, facilitate the introduction of high-quality rice resources, and accelerate local breeding programs. An in-depth analysis of the genetic and molecular mechanisms regulating rice flowering, as well as the exploration of the development of flowering regulation pathways, will enable breeders to provide more targeted and practical strategies for selecting the most suitable materials for local cultivation. Furthermore, these studies will provide optimal forms and combinations of flowering gene alleles for future molecular design breeding. CRISPR/Cas9 can regulate gene expression, creating new alleles that do not exist in nature and offering broad application prospects in future breeding of breakthrough and diversified rice varieties.

Recently, the CRISPR/Cas9 gene-editing system has become increasingly advanced. Compared to previous generations of gene-editing technology, the CRISPR/Cas9 system offers advantages such as more accessible construction, lower cost, and more straightforward operation. However, CRISPR/Cas9 has flaws, and its high off-target rate has been a subject of criticism. In this study, the *Badh2* gene target experienced off-target effects. To address this issue, Fu et al. developed a series of progressively shorter guide RNAs (gRNAs) for the *EGFP* reporter gene, including 15, 17, 19, or 20 complementary nucleotides. gRNAs with 17 or 18 complementary nucleotides could function at the intended target sites, reducing the off-target effects of paired Cas9 endonucleases without compromising the efficiency of targeted genome editing [38]. Research has shown that gRNAs can direct Cas9 for on-target editing via base pairing with “seed sequences” located 8–12 nucleotides upstream of the protospacer adjacent motif (PAM). Therefore, appropriately lowering the length of the gRNA by 1–3 nucleotides can effectively reduce the off-target probability without impacting its targeting ability [39]. In addition, we selected the physiological races 17-4 and Rb22 of *Magnaporthe oryzae*, known to be susceptible to varieties like ZH11 and ZH8015, and inoculated a mixture of these races into the JH212 variety [40]. However, the final plants did not get the relevant phenotypes, leading to the failure of the improvement of blast resistance. There are several possible reasons for this phenomenon; firstly, JH212 itself is inherently a cultivar with robust disease resistance and the physiological races of *Magnaporthe oryzae* we chose may exhibit sensitivity specifically to ZH11 and ZH8015, but JH212 is immune to these physiological races; secondly, the *pi21* gene mutation may have an insufficient genetic impact, resulting in the absence of related phenotypic expressions following the inoculation of the physiological races of *Magnaporthe oryzae* into these three lines. In our future research, we will select multiple physiological races of rice blast fungus to identify the resistance of JH212 to rice blast. Finally, we will compare the resistance of JH212 with gene-edited lines using the pathogenic physiological race with pathogenicity of JH212 for inoculation, and further evaluate the enhancement of rice blast resistance by mutations at the *Pi21* locus through experiments such as the expression of resistance related genes. In conclusion, despite certain limitations, CRISPR/Cas9 has attracted increasing attention from researchers due to its unique ability to modify genes precisely. This has prompted in-depth studies to overcome its challenges and examine its full potential. The three homozygous mutant lines obtained in this study, namely JH-C15, JH-C18, and JH-C31, and especially the last two, could serve as potential resources for expanding the promotion and application areas of JH212.

## 4. Materials and Methods

### 4.1. Experimental Materials and Planting

In this study, JH212 served as the recipient material for genetic transformation. Wild-type and transgenic plants from generations T_0_ and T_2_ were cultivated at the transgenic experimental base of the China Rice Research Institute. T_1_ plants were grown in Hainan during winter, receiving conventional water and fertilization practices. The expression vectors pC1300-Cas9 and SK-gRNA intermediate vectors were supplied by the laboratory of Wang Kejian at the China Rice Research Institute.

### 4.2. gRNA Target Site Design

The CRISPR-P website (http://skl.scau.edu.cn/, accessed on 25 July 2021) was utilized to design gRNA target sites. Primer sequences with high scores and proximity to the ATG start site were selected. Gene sequences for *Hd2*, *Ghd7*, *DTH8*, *Pi21*, and *Badh2* were inputted into the CRISPR-P website to facilitate target site design. The CRISPR/Cas9 system was used to recognize approximately 20 bases upstream of the PAM to identify suitable target sites. After identifying the target sites, we added the GGCA sticky end to the front and the CCCA sticky end to the back of the target site sequence. The specific primer sequences are detailed in Table 1. These five target sites were positioned close to the ATG start codon to facilitate the functional loss of the target genes. Lastly, the specificity of the target sites was confirmed by BLAST analysis of the rice genome using RAP DB (https://rapdb.dna.affrc.go.jp/, accessed on 25 July 2021).

### 4.3. Construction of CRISPR/Cas9 Expression Vector

Five genes were targeted for knockout using the CRISPR/Cas9 system. The successfully constructed expression vectors were introduced into Agrobacterium through homotetramer ligation. Primers Hd2-g-F/R, Ghd7-g-F/R, DTH8-g-F/R, Pi21-g-F/R, and Badh2-g-F/R (Appendix A) were synthesized based on the target site sequences. These primers were mixed in equal amounts, denatured, and annealed to form fragments with sticky ends. The sgRNA fragments for *Hd2*, *Ghd7*, *DTH8*, and *Pi21* were digested with *Kpn* I/*Xho* I, *Sal* I/*Xba* I, *Nhe* I/*BamH* I, and *Kpn* I/*Bgl* II, respectively, and then assembled into an intermediate vector. Subsequently, the two sgRNA fragments were combined into an intermediate vector containing all five sgRNAs. Finally, the intermediate vector containing the five sgRNAs was digested with *Kpn*I/*Bgl*II and ligated into the final expression vector pC1300-Cas9 (*Kpn*I/*BamH*I double digestion). The constructed CRISPR/Cas9-2 expression vector was used for the genetic transformation of the *japonica* rice variety JH212 by Wuhan BioRun biosciences Co., Ltd., Wuhan, China.

### 4.4. Acquisition of T0 Generation Positive Strains and Identification of Target Sites

The vector CRISPR/Cas9-2 was transferred to the JH212 rice callus using the Agrobacterium-mediated method, and hygromycin was applied for screening. At the full tillering stage, leaf DNA from the gene-edited rice plants was extracted using the CTAB method. Positive gene-edited plants were identified by PCR using primers Cas9-F/Cas9-R specific for the thaumatin resistance gene in the vector. Next, primers flanking the target sequences of genes *Hd2*, *Ghd7*, *DTH8*, *Pi21*, and *Badh2* were designed to amplify target bands from the T0-positive plants. The PCR products were then sequenced by Hangzhou Qingke Biotech Co. (Hangzhou, China), using the genome sequences of corresponding wild-type materials as references (Appendix A). Sequencing results were compared with the wild-type sequences. A single peak in the sequencing results indicated a homozygous mutation, while a double peak indicated a heterozygous mutation, which required further isolation and verification.

### 4.5. Screening for GM-Free Components in T1 and T2 Generations

Upon analyzing mutations at the editing site of transgenic plants in the T0 generation, it was observed that only a small proportion had homozygous mutations, with most showing heterozygous mutations. To obtain homozygous mutant plants without the Cas9 carrier, we employed self-cross segregation to remove the carrier from the transgenic plants. Leaf samples from T1 and T2 generation mutant plants were collected, and genomic DNA was extracted using the CTAB method. Genomic sequences of the mutant plants were amplified using vector-specific primers Hyg-F/Hyg-R and Cas9-F/Cas9-R. The amplified products were analyzed using 1% agarose gel electrophoresis. If both primers failed to amplify a band, the plant was identified as a positive mutant lacking exogenous transgene components.

### 4.6. qRT-PCR Analysis of Transgenic Plants

Total RNA was extracted from the leaves of JH212 and the homozygous mutant lines at flowering stege using the RNAprep homozygous Plant Kit. First strand cDNA was synthesized with the ReverTra Aceq PCR RT Master Mix with gDNA Remover (ToYoBo, Osaka, Japan), following the manufacturer’s instructions. The expression levels of *Hd2*, *Ghd7*, *DTH8*, and *Pi21* were analyzed in both wild-type and mutants using real-time fluorescence quantitative PCR (qRT-PCR) (Appendix A). The qRT-PCR protocol involved pre-denaturation at 95 °C for 30 s, followed by denaturation at 95 °C for 5 s, annealing at 58 °C for 30 s, and extension at 72 °C for 15 s, repeated for 35 cycles. The primers for quantitative PCR are shown in Table 2. The *UBQ* (accession number *LOC_Os03g13170*) served as the internal reference gene. Relative gene expression was calculated using the 2^-∆∆Ct^ method.

### 4.7. Mutant Phenotype Analysis

Phenotypic investigations were conducted on transgenic plants in the T_2_ generation, focusing on plant height and flowering period. The JH212 and mutant lines were planted under the same growing environment and growing conditions with consistent field water and fertilization management. For the flowering period, the measurement was taken 2 cm from the leaf sheath of the main spike of each plant, while plant height was measured as the distance from the top of the tallest spike to the soil surface after full irrigation. The data were recorded daily. To verify the effect of gene editing on the flowering period, a *t*-test was employed to compare wild-type rice materials and different mutants. When the JH212 and mutant lines were fully matured, 10 plants of each (excluding the side rows) were selected and examined for the grain length and width, the number of tillers, the yield per plant, and the 1000-grainweight. The data were analyzed using *t*-test, * and ** indicate significance at *p* ≤ 0.05 and *p* ≤ 0.01 levels, respectively. The data represent mean ± standard deviation.

## 5. Conclusions

In this study, we edited a specially designed Cas9 multi gene-editing vector and transformed three flowering-stage genes, *Hd2*, *Ghd7*, and *DTH8*, into the high quality *japonica* rice variety, JH212. This approach enabled us to produce progeny with different flowering stages and various genotypes, which were ideal materials for not only deep research on the relationship between genotypes and the flowering stage but also the creation of early maturing CMS (cytoplasmic male sterility, CMS) lines. Finally, we obtained three homozygous mutant lines, namely JH-C15, JH-C18, and JH-C31, which exhibited significantly earlier flowering without significant production loss. Our research provided new germplasm resources for the widespread cultivation of high-quality *japonica* rice varieties and offers significant support for future breeding initiatives focused on short-growth-period rice varieties.

## Figures and Tables

**Figure 1 plants-13-02166-f001:**
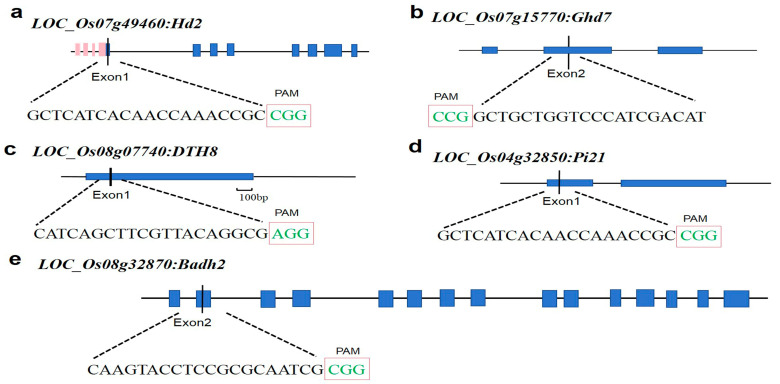
Target site location data. (**a**–**e**) correspond to *Hd2*, *Ghd7*, *DTH8*, *Pi21*, and Bad*h2*, respectively. The letters in the red box indicate the PAM sequence.

**Figure 2 plants-13-02166-f002:**
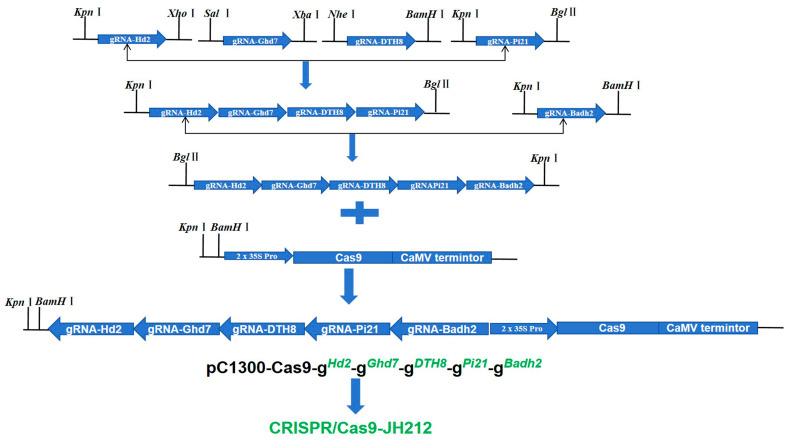
Expression vector construction procedure.

**Figure 3 plants-13-02166-f003:**
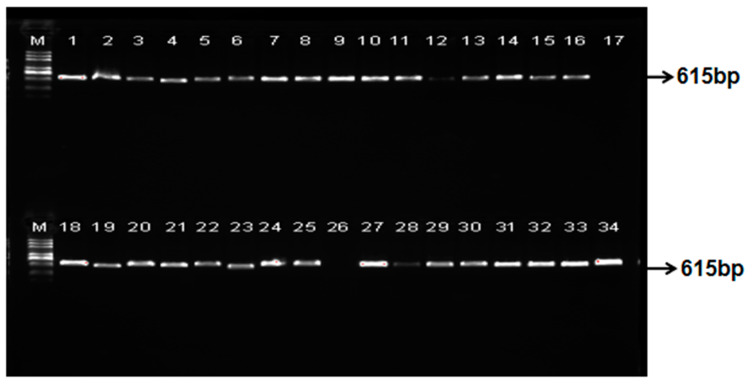
PCR screening T_0_ generation positive strains. M, 2K DNA marker; 1–34 is the test line number.

**Figure 4 plants-13-02166-f004:**
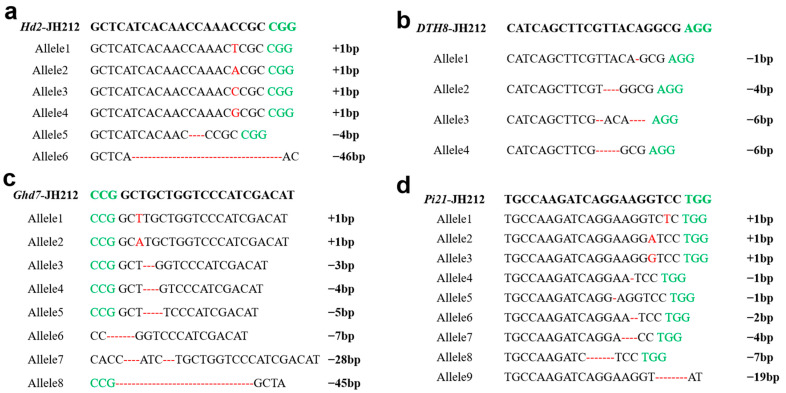
The mutant genotype of the positive T_1_ plants tested. The green letters indicate the PAM sequence; the red letters and hyphens indicate base insertions and missing bases, respectively. (**a**) The six mutation types of *Hd2*; (**b**) the four mutation types of *DTH8*; (**c**) the eight mutation types of *Ghd7*; (**d**) the nine mutation types of *Pi21*.

**Figure 5 plants-13-02166-f005:**
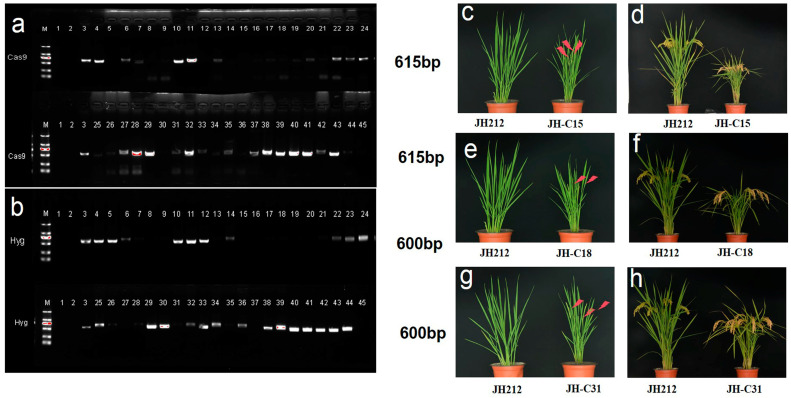
PCR screening of T_2_ generation mutant lines without transgenic components and the phenotype of the three selected homozygotes lines. (**a**,**b**) PCR screening of the Cas9 vector region and *Hygromycin* region in the single plants sampled from the three selected homozygotes lines. M, 2K DNA marker; 1–3, are negative control, distilled water, positive control, 4–16 are strains of JH-C15, 17–30 are strains of JH-C18, a31–a45 and 31–45 are strains of JH-C31. (**c**,**e**,**g**) the phenotype of JH-C15, JH-C18, and JH-C31 at flowering stage; (**d**,**f**,**h**) the phenotype of JH-C15, JH-C18, and JH-C31 at harvest stage.

**Figure 6 plants-13-02166-f006:**
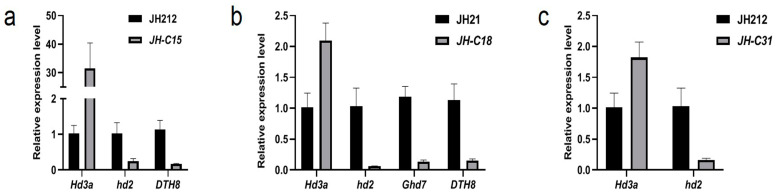
qPCR analysis of flowering-related genes in the selected homozygotes lines. (**a**–**c**) Relative expression levels of flowering stage related genes in the homozygous mutant lines JH-C15, JH-C18 and JH-C31, respectively.

**Figure 7 plants-13-02166-f007:**
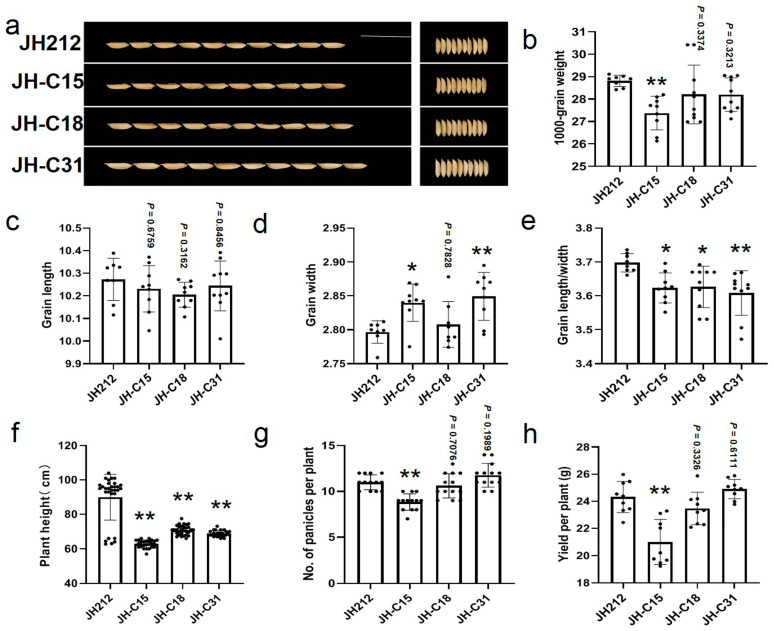
Analysis of grain shape and agronomic traits of the selected homozygotes lines. (**a**–**h**), grain appearance and shape, thousand-grain weight, length mean, width mean, length and width mean ratio, effective tiller number, and yield per plant. The data were analyzed using *t*-test, * and ** indicate significance at *p* ≤ 0.05 and *p* ≤ 0.01 levels, respectively. The data represent mean ± standard deviation.

**Table 1 plants-13-02166-t001:** Off-target frequency assessment.

Target	Position	Strand	GC (%)	Region	Potential Off-Target Sites (Max Score)	Pairing with sg RNA (>=8 nt)
*Hd2*	738–757	+	55	CDS1	0.074	None
*Ghd7*	366–385	+	60	CDS2	0.25	None
*DTH8*	986–1005	+	55	CDS1	0.238	None
*Pi21*	503–522	+	55	CDS1	0.133	None
*Badh2*	499–518	+	60	CDS2	0.083	None

**Table 2 plants-13-02166-t002:** Mutation frequency of positive plants in T_0_ generation.

Gene	No. of Plants	Number of Mutant Strains	Mutation Rate
*Hd2*	30	26/30	86.66%
*Ghd7*	30	25/30	83.33%
*DTH8*	30	25/30	83.33%
*Pi21*	30	26/30	86.66%

**Table 3 plants-13-02166-t003:** Genotype combination analysis of T_1_ homozygous lines.

Line No.\Gene	*Hd2*	*Ghd7*	*DTH8*	*Pi21*	*Badh2*	Flowering after Sowing (Days)
JH212	WT	WT	WT	WT	WT	88.8 ± 3.5
JH-C15	Allele1: +1bp	WT	Allele1: −4 bp	Allele3: +1 bp	WT	57.38 ± 2.75 **
JH-C17	Allele1: +1 bp	Allele1: +1 bp	WT	Allele6: −2 bp	WT	53.22 ± 2.75 **
JH-C18	Allele6: −46 bp	Allele1: +1 bp	Allele1: −1 bp	Allele1: +1 bp	WT	61.4 ± 2.25 **
JH-C19	Allele2: +1 bp	Allele7: −28 bp	WT	Allele2: +1 bp	WT	56.55 ± 3.25 **
JH-C22	Allele2: +1 bp	Allele1: +1 bp	WT	Allele9: −19 bp	WT	56.55 ± 4.25 **
JH-C23	Allele3: +1 bp	Allele5: −5 bp	WT	Allele2: +1 bp	WT	50.5 ± 2.75 **
JH-C24	Allele3: +1 bp	Allele4: −4 bp	WT	Allele9: −19 bp	WT	55.5 ± 3.25 **
JH-C26	Allele1: +1 bp	Allele5: +1 bp	WT	Allele6: −2 bp	WT	50.5 ± 2.25 **
JH-C27-5	Allele2: +1 bp	WT	WT	Allele7: −4 bp	WT	51.55 ± 4.25 **
JH-C27-2	WT	Allele7: −28 bp	WT	Allele7: −4 bp	WT	52.55 ± 3.25 **
JH-C28-5	Allele4: +1 bp	Allele2: +1 bp	WT	Allele8: −7 bp	WT	60.55 ± 2.75 **
JH-C31	Allele5: −5 bp	WT	WT	Allele4: −1 bp	WT	59.07 ± 2.75 **
JH-C32	Allele2: +1 bp	WT	WT	Allele4: −1 bp	WT	60.55 ± 3.25 **
JH-C34	Allele5: −5 bp	Allele5: −5 bp	WT	Allele5: −1 bp	WT	63.55 ± 2.75 **
JH-C37	Allele3: +1 bp	Allele3: −3 bp	WT	Allele2: +1 bp	WT	59.55 ± 3.25 **
JH-C38	Allele2: +1 bp	Allele8: −45 bp	WT	Allele9: −19 bp	WT	55.5 ± 3.75 **
JH-C40	Allele4: +1 bp	Allele3: −3 bp	WT	Allele4: −1 bp	WT	60.55 ± 2.75 **

** indicate significance at *p* ≤ 0.01 levels.

## Data Availability

Data are contained within the article and Appendix A.

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
