# Peer review of "Improvement of Flowering Stage in *Japonica* Rice Variety Jiahe212 by Using CRISPR/Cas9 System"

_plants, 2024, doi:10.3390/plants13152166_

Round 1

Reviewer 1 Report

Comments and Suggestions for Authors

I have the following concerns regarding this manuscript.

1, The title of the manuscript seems not consistent with the content. There’s no test for the ecological adaptability of genetically modified rice. Also, the “multiplex genome editing technology” seems have not much difference with the ordinary CRISPR/Cas9 genome editing technology.

2, The figure legends should be updated with more information.

3, Line 83-86: “Delaying rice flowering under prolonged sunlight conditions produced increased plant height and number of grains per spike, leading to higher yields. Therefore, it is important to shorten the rice flowering period by editing Hd2, Ghd7, and DTH8, to genetically improve the flowering period of Geng rice variety.” This sentence seems not clear and may need to be rephrased.

4, Line 273-274: “Ehd1 promotes the expression of Hd3a and RFT1 under long sunlight conditions”. This sentence seems not accurate. Ehd1 promotes the expression of Hd3a and RFT1 under both long and short sunlight conditions.

5, Line 293: “Hence, it resulted in showing different agronomic traits”. This sentence may need to be rephrased.

Comments on the Quality of English Language

Minor English editing may improve the quality of the manuscript.

Author Response

Dear Editors and Reviewers:

Thank you for your letter and for the reviewers’ comments concerning our manuscript entitled

Application of Multiplex Genome Editing Technology on Targeted Improvement of Ecological Adaptability of the Geng Rice Variety Jiahe212” (plants-3027683). Those comments are all valuable and very helpful for revising and improving our manuscript, as well as the important guiding significance to our research. We have studied these comments carefully and have made the corrections which we hope meet with your approval. The revised portion are marked with tracked changes in the revised manuscript. The responses to the reviewers’ comments are as following:

Responses to the comments of Reviewer #1

Response: Thank you for your valuable comments. We have studied these comments carefully and have made the corrections which we hope meet with your approval. The revised portion are marked with tracked changes in the revised manuscript. The responses to the reviewers’ comments are as following:

1, The title of the manuscript seems not consistent with the content. There’s no test for the ecological adaptability of genetically modified rice. Also, the “multiplex genome editing technology” seems have not much difference with the ordinary CRISPR/Cas9 genome editing technology.

Response: Thank you for your valuable comments. Our research primarily focuses on the creation of breeding materials, encompassing traits such as heading date and resistance to rice blast disease. However, ecological adaptation is not merely a matter of trait differences, but a matter of validation or observation regarding whether specific species or genotypes can actually adapt to a wider range of ecosystems or ecological conditions due to changes in traits. Nevertheless, our study has not yet conducted experiments to verify changes in ecological adaptation, but only adjusted the growth period. Therefore, as pointed out by the editor and you, the title of this article does not align precisely with the research content. Based on your suggestion, we change the title of this article to " Improvement of Flowering Stage in the japonica Rice Variety Jiahe212 by Using CRISPR/Cas9 System"

The CRISPR/Cas9 system used in this study, which has been extensively validated in the Wang Kejian’s laboratory at CNRRI, demonstrates robust applicability. This study serves as a further validation and application of this system in Jiahe212.We have recognized the mistakes in the previous description of the system within the text, which has been rectified in both the text and the title and we hope our corrections can meet with your approval. Thank you again for your professional comments.

  1. The figure legends should be updated with more information.

Response:  Thank you very much for your important comments. We have made careful adjustments and added detailed information in the figures. Thanks again for your valuable comments.

  1. Line 83-86: “Delaying rice flowering under prolonged sunlight conditions produced increased plant height and number of grains per spike, leading to higher yields. Therefore, it is important to shorten the rice flowering period by editing Hd2, Ghd7, and DTH8, to genetically improve the flowering period of Geng rice variety.” This sentence seems not clear and may need to be rephrased.

Response: Thank you very much for your important comments. We have already corrected the sentence.

  1. Line 273-274: “Ehd1 promotes the expression of Hd3a and RFT1 under long sunlight conditions”. This sentence seems not accurate. Ehd1 promotes the expression of Hd3a and RFT1 under both long and short sunlight conditions.

Response: Thank you very much for your important comments. We have already corrected the sentence.

  1. Line 293: “Hence, it resulted in showing different agronomic traits”. This sentence may need to be rephrased.

Response: Thank you very much for your important comments. We have already corrected the sentence to

Reviewer 2 Report

Comments and Suggestions for Authors

As indicated in their title, the authors of this paper have used multiplex genome editing to influence flowering time in Geng rice with the hope of improving adaptability.  The paper is basically a progress report with the successful selection of three mutant lines that have earlier flowering times without significant production loss.  While I have no reason to question their data, the authors have made no attempt to experimentally test any of their suggestions regarding mechanisms underlying the changes nor have they tried breeding the new traits.  Accordingly, I feel this manuscript is too preliminary for publication.  The technology used to produce the new traits is not new or unique to this study and biologically significant findings are not reported.  

Comments on the Quality of English Language

No additional comments.

Author Response

Thank you for your valuable comments. In this study, we mainly focused on the creation of breeding materials and the agronomic traits of gene edited lines, so we hardly paid attention to mechanisms underlying of rice flowering. You have provided us with excellent ideas for improving and conducting in-depth research on these gene editing materials, as well as future our research directions. We will follow up with your suggestion to design relevant experiments for our teachers and junior brothers and sisters in our further research.

Sincerely, we believe that gene editing technology is a desirable method for crop improvement, and multiple gene editing vectors are even available. In fact, the selected lines showed various early flowering were really obtained in this study. Our next step in utilizing these materials to further develop three line CMS lines is still a long way to go. We will fully consider and listen to your opinions, and comprehensively design future research.

Reviewer 3 Report

Comments and Suggestions for Authors

This is a well written and methodologically sounded manuscript. The study demonstrated a successful application of the multiplex genome editing technology CRISPR/Cas9 in creating potential resources for targeted improvement of widely adaptabilities of the Geng rice varieties. In general:

1. The context of the study is provided, and the significance of the findings is supported.

2. The statistical analysis, the interpretation and discussion of the results are appropriate.

3. The paper tells a convincing story and is a good addition to the literature.

 Minor changes suggested:

1.       The authors need to follow the standard format of the MDIP journal which is logical and easier to follow. Readers expect to read the research materials and methodologies before exploring the results.

2.       How did the authors collect data of each agronomic trait, please make it clearer? Measuring parameters need to be clarified in text and figures (figure 8, page 10).    

3.       What recommendations do the authors want to make for the use of the tools in modifying targeted genes in other cereal crops?

Author Response

Dear Editors and Reviewers:

Thank you for your letter and for the reviewers’ comments concerning our manuscript entitled

Application of Multiplex Genome Editing Technology on Targeted Improvement of Ecological Adaptability of the Geng Rice Variety Jiahe212” (plants-3027683). Those comments are all valuable and very helpful for revising and improving our manuscript, as well as the important guiding significance to our research. We have studied these comments carefully and have made the corrections which we hope meet with your approval. The revised portion are marked with tracked changes in the revised manuscript. The responses to the reviewers’ comments are as following:

Responses to the comments of Reviewer #3

Major comments:

  1. The authors need to follow the standard format of the MDIP journal which is logical and easier to follow. Readers expect to read the research materials and methodologies before exploring the results.

Response: Thank you very much for your important comments. We have made careful adjustments following the standard format of MDPI journals. Thanks again for your valuable comments.

  1. How did the authors collect data of each agronomic trait, please make it clearer? Measuring parameters need to be clarified in text and figures (figure 8, page 10). 

Response: Thank you very much for your professional comments. We have added the measurement parameters at Figure 8 and described in detail of the collection methods for the data obtained for each of the agronomical traits in the revised manuscript. Thank you again for your professional advice!

  1. What recommendations do the authors want to make for the use of the tools in modifying targeted genes in other cereal crops.

Response: Thank you very much for your professional comments. The expansion of the CRISPR technology enables a wide range of genomic manipulations such as gene knockouts, deletions, base editing, insertions or substitutions, and RNA editing, providing a highly efficient and precise tool as well as new avenues and possibilities for genetically oriented crop improvement. Genome editing help researchers to precisely target and edit desired genes to obtain improved lines with high productivity, nutritional value and tolerance to biotic and abiotic stresses. This has been widely used in cereal crops, we have added several successful cases of gene editing in crop genetic improvement in the revised manuscript which we hope meet with your approval.

Round 2

Reviewer 2 Report

Comments and Suggestions for Authors

The authors have made several text changes but made no attempt to address my concerns with additional data or experiments.  Accordingly, my concerns remain as stated and I must recommend rejection.

Comments on the Quality of English Language

No additional comments.

Author Response

Dear Reviewer:

Thank you for your professional comments concerning our manuscript entitled “Application of Multiplex Genome Editing Technology on Targeted Improvement of Ecological Adaptability of the Geng Rice Variety Jiahe212” (plants-3027683). Those comments are all valuable and very helpful for revising and improving our manuscript, as well as the important guiding significance to our research. We have studied these comments carefully and have made some corrections which we hope meet with your approval.

 In fact, we do have some difficulties, and we earnestly request your understanding. Previously, we observed that the primary focus of this special issue is on genetic improvement research related to breeding materials. Given our dedication to crop genetic improvement research, we believe our study aligns with the submission requirements of the special issue and have submitted our findings accordingly.

Indeed, as you pointed out, we need to provide additional experimental data with biological significance. The molecular mechanism of heading stage is extremely complex, and gene editing technology is also a beneficial tool for analyzing the gene network of heading stage. We should have taken this into consideration when designing the project, but as we are all breeders, we have overlooked this aspect. Currently, we are encountering several challenges which we must explain to you.

Firstly, the primary author graduated in June this year, posing obstacles in conducting further experiments.

      Secondly, the material we obtained concerning this material was intended to be used primarily as breeding material, mainly for the creation of CMS lines, and therefore relevant follow-up experiments in a biological sense were not assigned and carried out.

      Additionally, the supplementary experiments encompass diverse areas like rice heading date, blast resistance identification, and agronomic trait analysis. However, since the improved strains that we have obtained so far are at the seedling stage, completing the required experiments within a short timeframe is not feasible.

Despite the aforementioned challenges, our study maintains a clear objective for crop genetic improvement, namely, obtaining improved JH212 lines. Given the current experimental findings, we have attained our preset experimental objectives and successfully achieved the targeted enhancement of rice heading date. Consequently, we believe our study continues to meet the submission criteria for this special issue.

     We sincerely appreciate your valuable time in reviewing our manuscript and providing professional advice despite your busy schedule. Thank you also once again for giving us the opportunity to explain ourselves, and we will endeavour to overcome the existing difficulties and strive to achieve better results in our future research.

     Ultimately, we aspire for the publication of our manuscript in "Plants". Our findings will demonstrate the breadth of coverage exhibited by the journal across diverse fields, Thereby highlighting the comprehensive and exceptional quality of "Plants" as a journal. Thank you again for giving us the opportunity to explain.
    Best wishes to you!

Lianling Sun